# Metabolic and circadian inputs encode anticipatory biogenesis of hepatic fed microRNAs

Sandra Usha Satheesan[1], Shreyam Chowdhury[1], Ullas Kolthur-Seetharam[1,2]

**Starvation and refeeding are mostly unanticipated in the wild in terms of duration, frequency, and nutritional value of the refed state. Notwithstanding this, organisms mount efficient and reproducible responses to restore metabolic homeostasis. Hence, it is intuitive to invoke expectant molecular mechanisms that build anticipatory responses to enable physiological toggling during fed-fast cycles. In this regard, we report anticipatory biogenesis of oscillatory hepatic microRNAs that peak during a fed state and inhibit starvation-responsive genes. Our results clearly demonstrate that the levels of primary and precursor microRNA transcripts increase during a fasting state, in anticipation of a fed response. We delineate the importance of both metabolic and circadian cues in orchestrating hepatic fed microRNA homeostasis in a physiological setting. Besides illustrating metabo-endocrine control, our findings provide a mechanistic basis for the overarching influence of starvation on anticipatory biogenesis. Importantly, by using pharmacological agents that are widely used in clinics, we point out the high potential of interventions to restore homeostasis of hepatic microRNAs, whose deregulated expression is otherwise well established to cause metabolic diseases.**

## Introduction

Fed-fast cycles and the associated anabolic-catabolic transitions are vital for physiological homeostasis, and inefficient toggling is known to cause metabolic diseases and accelerated aging (Ribarič, 2012; Suliga et al, 2015; Geisler et al, 2016; Petersen et al, 2017; Paoli et al, 2019). Evolutionarily, these transitions are coupled with circadian oscillations and dictated by light-dark and sleep/inactive-wake/active cycles (Gerhart-Hines & Lazar, 2015; Pickel & Sung, 2020). Physiological changes that accompany fed-fast and circadian cycles are dependent upon gene expression cascades, especially in central metabolic tissues such as the liver (Jitrapakdee, 2012; Bideyan et al, 2021). Even though decades of work have illustrated

the congruence of nutrient and circadian inputs for gene transcription, if/how they impact post-transcriptional regulation of metabolic gene programs is relatively less understood.

MicroRNA (miR)-dependent gene expression control is well established and particularly important for mediating dynamic changes in mRNA translation, storage, and turnover (Wilczynska & Bushell, 2015; O'Brien et al, 2018; Dexheimer & Cochella, 2020). We and others have demonstrated the significant role of miR-mediated regulation of gene expression in orchestrating physiological responses including during fed-fast cycles (Hu et al, 2012; Rottiers & Näär, 2012; Maniyadath et al, 2019; LaPierre et al, 2022). In addition to specific miR signatures that are associated with fed and fasted states in the liver (Na et al, 2009; Vollmers et al, 2012; Ji et al, 2023), hepatic miRs have also been shown to impinge on distant organ systems (Sung et al, 2018; Liu et al, 2020; Ji et al, 2021). However, metabolic inputs/signals that are essential for miR homeostasis, from biogenesis and processing to degradation and secretion, are largely unknown. This is likely the case since, as detailed later, activities of drosha and dicer have been shown to be regulated in other physiological contexts (Davis & Hata, 2009; Creugny et al, 2018). Moreover, given that fed-fast cycles are inherently linked to circadian rhythm, there is a paucity of information vis-à-vis phenomenological and mechanistic underpinnings of hepatic miR biogenesis and processing.

Decades of work have demonstrated that organismal behaviour and physiology, which are associated with feeding and fasting, involve anticipation, and are largely mediated by cephalic responses (Zafra et al, 2006; Power & Schulkin, 2008; Smeets et al, 2010). Cephalic responses are conditioned hormonal and metabolic adaptations, and perturbing such anticipatory responses has been shown to have detrimental effects (Berthoud et al, 1980; Ahrén & Holst, 2001; Zafra et al, 2006). For example, blocking the cephalic phase of insulin secretion causes dysregulated glucose homeostasis (Berthoud et al, 1980; Ahrén & Holst, 2001). Whereas the current understanding of fed-fast anticipation is dominated by cephalic mechanisms that operate at an organismal level, it is still unclear if anticipatory cellular-level mechanisms facilitate effective switching between anabolic and catabolic states. Furthermore, whether such cellular/molecular anticipatory mechanisms are tuned by neuroendocrine and metabolic signals remains unknown.

[1]Department of Biological Sciences, Tata Institute of Fundamental Research, Mumbai, India    [2]Tata Institute of Fundamental Research- Hyderabad (TIFR-H), Hyderabad, India

Correspondence: ullas@tifr.res.in

It is interesting to note that the hepatic fed microRNAs that we previously characterized as regulators of hepatic functions and organismal physiology exhibited anticipatory biogenesis (Maniyadath et al, 2019). Specifically, the dynamic post-transcriptional convergent and additive control exerted by the fed miR network entailed the transcription of these microRNAs in a fasted/starvation state. Among these microRNAs, let-7i, miR-221, miR-222, and miR-204 have now emerged as key players in various liver pathological conditions including fibrosis and hepatocellular carcinoma (Chen et al, 2014; Luo et al, 2017; Song et al, 2017; Di Martino et al, 2022; Kim et al, 2022). Therefore, identifying the upstream mechanisms that govern their biogenesis will not only provide fundamental insights into molecular anticipation but also possibly open up new avenues for therapeutic interventions.

It is intriguing to note that despite the high volume of information vis-à-vis miR-dependent regulation of cellular and organismal functions (Ebert & Sharp, 2012; Dexheimer & Cochella, 2020), including metabolism and physiology (Rottiers & Näär, 2012; Vienberg et al, 2017), our understanding of mechanisms that dictate miR homeostasis is still poor. Seminal studies have illustrated that Ras/MAPK and TGF-/BMP signalling pathways play a major role as regulators of microRNA biogenesis (Hata & Davis, 2009; Kent et al, 2010; Saj & Lai, 2011). Even though studies have also unravelled specific transcription factors that are necessary for miR expression (Davis & Hata, 2009; Saj & Lai, 2011; Creugny et al, 2018), little is known about differential and/or layered control of miR processing that together determine biogenesis. For example, the expression and activity of drosha and dicer have been shown to be regulated in response to stress (Davis & Hata, 2009; Beezhold et al, 2010; Creugny et al, 2018). Albeit these reports, in conjunction with emerging evidence on RNA methylation, portend layered regulation of miR homeostasis, the physiological significance of such regulation, particularly in the context of hepatic fed-fast transitions, is unknown as yet. Moreover, although hepatocyte nuclear factor $4\alpha$ (HNF4$\alpha$) and peroxisome proliferator-activated receptor $\alpha$ (PPAR$\alpha$) have been shown to regulate transcription of miR-genes in hepatocytes (Li et al, 2011; Cheng et al, 2017), it will be important to investigate if these are orchestrated by endocrine and metabolic signals to create molecular anticipation vis-à-vis fed miR biogenesis.

In the present study, we clearly illustrate the anticipatory expression of hepatic fed microRNAs during a starvation state. We provide phenomenological and mechanistic underpinnings to delineate the contributions of fed-fast and circadian inputs in eliciting this anticipatory expression, besides providing the complex regulation of primary, precursor, and mature microRNA transcripts in the liver.

# Results

### Diurnal oscillation in hepatic miR biogenesis

An earlier report from our group discovered an oscillatory or dynamically reversible pattern of miRs responsive to metabolic cues (Maniyadath et al, 2019). Moreover, our findings indicated anticipatory expression of hepatic fed miRs that was crucial for rapid silencing of the expression of starvation-dependent genes during fed-fast-refed cycles. The anticipatory and oscillatory expression, which was hitherto unknown, motivated us to unravel upstream molecular/ physiological mechanisms using specific hepatic miRs, let-7i, miR-221, miR-222, and miR-204 that showed the most robust changes during fed-starved-refed states (Fig 1A). Therefore, to address the diurnal rhythmicity of hepatic fed microRNAs let-7i, miR-221, miR-222, and miR-204, we harvested livers from ad libitum–fed mice sacrificed at 4-h intervals over the course of a 24-h light-dark cycle. We found that these microRNAs oscillate in a diurnal fashion (Fig 1B) and correlate well with the circadian-dependent expression of Cry1 and Per2 (Fig S1A). Notably, their expression progressively decreased in the inactive light phase (ZT4-ZT12) and increased in the active dark phase (ZT16-ZT24) (Fig 1B). These time points corresponded with starved and fed states, as indicated by the respiratory exchange ratio (RER) under ad libitum conditions (Fig S1B). The changes observed here were consistent with our earlier work (Maniyadath et al, 2019) and inversely correlated with the abundance of target starvation-dependent transcripts such as Sirt1, Ppargc1a, Acadl, Acadm, and Tfam (Fig S1C).

The anticipatory expression of microRNAs (Maniyadath et al, 2019) led us to assess miR biogenesis in ad libitum–fed, 24-h–starved, and 6-h–refed mice. As expected, we observed an increase in primary (pri) and precursor (pre) transcripts of these microRNAs in 24 h starvation, which was reduced or brought down to basal level upon refeeding (Fig 1C). This prompted us to investigate the oscillation of the primary and precursor transcripts across ZT time points in ad libitum–fed mice. Interestingly, both pri- and pre-transcripts of fed microRNAs were elevated in the starvation/ inactive phase in a miR-specific manner (Fig 1D and E). For example, levels of pri-let-7i and pri-miR-204 exhibited a progressive increase from ZT4 onwards and peaked around ZT8-ZT12 (Fig 1D). Whereas pri-miR-221 and pri-miR-222 also showed elevated levels in the light/starvation phase, their maximal expression was seen earlier at ZT4 (Fig 1D). Regardless of such differences in pri-miRs, pre-miRs showed similar patterns of induction during the light/ starvation phase, with a peak around ZT8-ZT12 (Fig 1E). Together, these results demonstrated that the pri-, pre-, and mature transcripts of microRNAs let-7i, miR-221, miR-222, and miR-204 showed diurnal rhythmicity. Furthermore, they indicated a possible association between circadian and fed-fast cues in regulating miR homeostasis, that is, in addition to validating the anticipatory nature of miR biogenesis.

### Temporally restricted feeding paradigms rewire miR biogenesis in the liver

It is well established that circadian oscillations and fed-fast cycles are intrinsically coupled (Pickel & Sung, 2020). Specifically in the liver, nutrient/metabolic inputs have been shown to tune circadian-dependent molecular rhythmicity and thus exert control over the peripheral clock (Lamia et al, 2008; Vollmers et al, 2009; Greenwell et al, 2019). Given this and based on the results presented above, we wanted to delineate between fed-fast and circadian dependence of anticipatory pri-miR and pre-miR expression. Towards this, we first used a time-restricted feeding paradigm where food was made

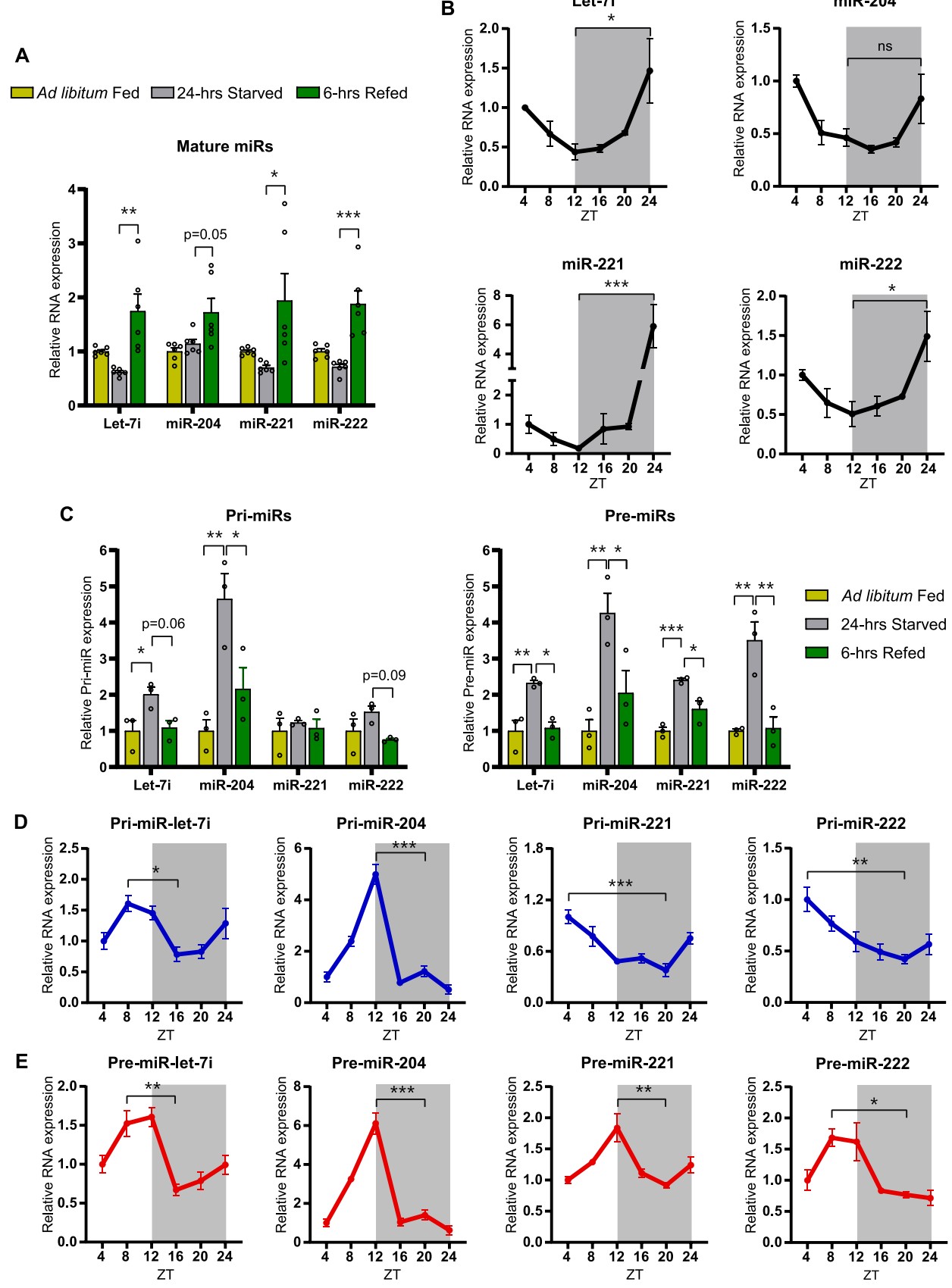

available either in the dark phase (ZT12–ZT24) (dark-fed) or in the light phase (ZT0–ZT12) (light-fed) (Fig 2A). Assaying for RER and *Cry1* expression, following the 14-d entrainment period, clearly showed the switch in whole-body energetics and inverted circadian rhythm in the liver (Fig 2B–D). Consistent with our hypothesis, we indeed found the level of mature miRs to be reversed in the light-fed and dark-fed cohorts, with peak expression in fed states (Fig S2) and as indicated above, anti-correlated with the target mRNAs (Fig 2D).

Not surprisingly, the expression profiles of pri-, pre-, and mature microRNA (Figs 2E and F and S2) transcripts in the dark-fed group were similar to those of ad libitum–fed mice and agreed well with fed-fast dependence. The only notable difference that we observed between ad libitum and dark-fed groups was for pri-miR-221 and pri-miR-222, which showed an additional peak at ZT16 that was absent in the ad libitum–fed mice (Fig 2E). Importantly, whereas pri-let-7i and pri-miR-204 showed complete inversion in expression, it was subdued for pri-miR-221 and pri-miR-222 when mice were fed during the light phase (Fig 2E). Irrespective of this, it was interesting to observe that precursor transcripts of all the miRs assayed demonstrated retroverted expression (Fig 2F). Moreover, the maximal abundance of pre-miRs was seen at ZT12 and ZT24 in a paradigm-specific manner (Fig 2F), interestingly at time points that correlate with peak starvation and transition to dark or light phases, respectively.

## Circadian independent regulation of let-7i/miR-204 but not miR-221/-222

Because both light–dark and fed–fast inputs are zeitgebers (Pickel & Sung, 2020), we next wanted to check if miR biogenesis was independent of circadian/light–dark inputs. Towards this, we used paradigms that perturbed both light–dark and fed–fast cycles (as detailed in the Materials and Methods section) and henceforth referred to as AL-LD (ad libitum light-dark paradigm) and S-LD (starvation light-dark paradigm) (Fig 3A). Specifically, to assess if anticipatory biogenesis was responsive to the duration/extent of starvation, we removed the feed at ZT0 and harvested livers from mice euthanized at 4 h intervals, as indicated (Fig 3A). Whereas we found that the expression of clock and starvation-responsive metabolic genes were consistent with earlier reports (Fig S3A and B) (Vollmers et al, 2009; Kinouchi et al, 2018), the primary and precursor transcripts of let-7i and miR-204 had lost oscillatory expression in response to continuous starvation, unlike in the ad libitum–fed mice (Fig 3B and C). Notably, the decrease associated with the transition from the light to dark phase was absent (Fig 3B and C) and clearly hinted at transcriptional induction of let-7i and

mir-204 independent of circadian inputs. In contrast, the response of pri-/pre-221/-222 was distinct and indicated an interplay between both fed–fast and circadian cues in regulating their homeostasis (Fig 3B and C). Albeit intriguing, the patterns displayed by pre-miR-221 and pre-miR-222 were disparate despite having common primary transcript (Wurz et al, 2010).

To affirm these results and to provide conclusive evidence vis-à-vis dependence on either or both of the zeitgebers, mice housed in continuous dark cycle were subjected to starvation or fed ad libitum (S-DD and AL-DD, respectively), as indicated (Fig 3D). Interestingly, we found that the changes in expressions of the miRs assayed in AL-DD mice (Fig 3E and F) were nearly indistinguishable from AL-LD mice (Fig 3B and C). For example, primary and precursor transcripts of let-7i and miR-204 showed peak expression between ZT8 and ZT12, and a fall subsequently (Fig 3E and F). More importantly, levels of these pri- and pre-miRs in S-DD (Fig 3E and F) mice mirrored the pattern that was observed in S-LD (Fig 3B and C) mice. Together, these results clearly indicated that especially for let-7i and miR-204 starvation cues but not light dependent circadian inputs regulated their expression.

## Transcriptional and post-transcriptional processing control biogenesis of hepatic fed miRs

Having demonstrated fed–fast dependence of hepatic miR biogenesis, we further wanted to gain mechanistic insights into starvation-dependent increases of pri- and pre-miRNAs, and refed mediated up-regulation of mature miRs. Towards providing conclusive molecular evidence for starvation-dependent induction of miR biogenesis, which is primarily associated with transcription, we performed chromatin immunoprecipitation and assayed for the abundance of serine 2 phosphorylated RNA polymerase II (S2P-Pol II). Specifically, we wanted to quantify S2P-Pol II that is elongating, transcriptionally active RNA Pol II on gene bodies of hepatic miRs in ad libitum–fed and 24-h–starved mouse liver. As anticipated, we observed higher abundance of S2P-Pol II on let-7i, miR-204, and miR-221/222 genes in starved state when compared to the fed state (Figs 4A and S4A), clearly demonstrating the starvation-dependent anticipatory transcription of these miRs. Our previous study identified miR-99b to be one of the hepatic miRs that remained unchanged during fed–fast cycles, and therefore, not surprisingly, we observed no significant change in the abundance of Pol II on miR-99 locus (Fig S4A).

Expression levels and activity of DROSHA, DGCR8, and DICER have been shown to be regulated under various contexts (Davis & Hata, 2009; Beezhold et al, 2010; Ha & Kim, 2014; Creugny et al, 2018). Thus, we surmised a potential RNA processing machinery dependent

---

**Figure 1. Diurnal oscillation of anticipatory microRNA biogenesis in mouse liver.**
**(A)** Relative expression of mature miRNAs (miRs)—let-7i, miR-221, miR-222, and miR-204 in ad libitum–fed, 24-h–starved, and 6-h–refed mice. miR-transcript levels were normalized to 18S rRNA level in each case and plotted as fold change with respect to ad libitum fed condition (N = 2, n = 3). **(B)** Relative expression of mature miRs from the liver of ad libitum–fed mice at the indicated ZT (Zeitgeber time). miR-transcript levels were normalized to 18S rRNA level and plotted as fold change with respect to ZT4 (N = 2, n = 3). **(C)** Relative expression of primary miRNAs (pri-miRs) and precursor miRNAs (pre-miRs) in ad libitum–fed, 24-h–starved, and 6-h–refed mice. Pri- and pre-miR-transcript levels were normalized to *Actin* mRNA level and plotted as fold change with respect to ad libitum fed (N = 2, n = 3). **(D, E)** Relative expression of (D) Pri-miR and (E) Pre-miR transcripts from the liver of ad libitum–fed mice at the indicated ZT. Pri- and pre-miR-transcript levels were normalized to *Actin* mRNA level and plotted as fold change with respect to ZT4 (N = 2, n = 3). Data information: In (A, B, C, D, E), data are represented as mean ± SEM. Statistical significance was calculated using one-way ANOVA with Tukey's test for multiple comparisons between groups (*$P < 0.05$; **$P < 0.01$; ***$P < 0.001$).

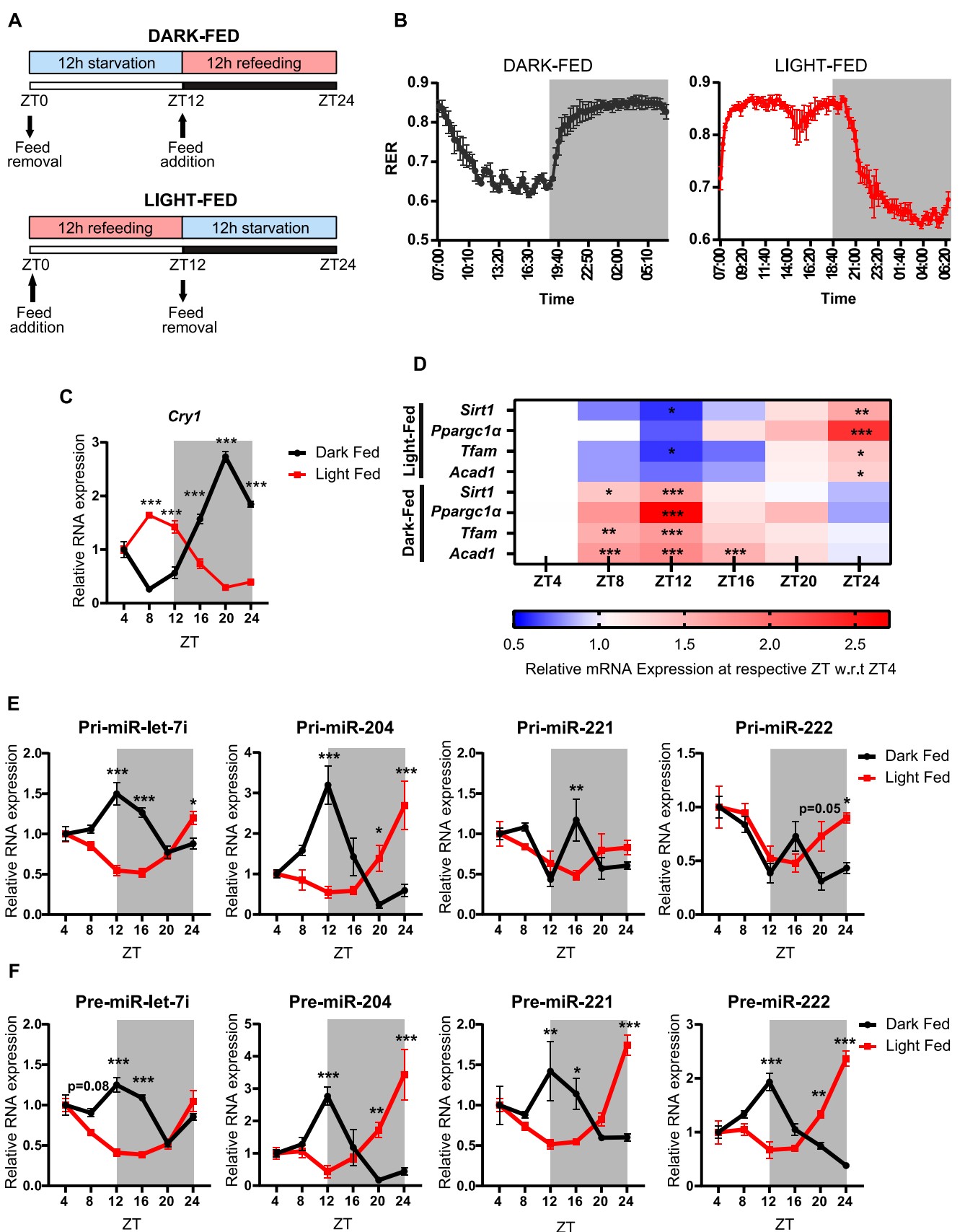

dynamic regulation of pri-/pre- and mature miRs under fasted and refed states, respectively. We assayed the protein levels of DROSHA and DICER in ad libitum–fed, 24-h–starved, and 6-h–refed states, as indicated (Figs 4B and C and S4B and C). Whereas DICER levels were robustly up-regulated in the refed state, they showed a marginal but significant decrease in response to starvation (Figs 4B and C and S4B and C). Notably, increased DICER in the refed state correlated well with the rapid increase in mature-miR levels during fasted-to-refed transitions. This prompted us to investigate if fed endocrine inputs, in the form of insulin, were necessary to regulate the levels of mature miRs. Treating primary hepatocytes with insulin, as detailed in the Materials and Methods section, led to an increase in mature miR-221 and miR-222 (Fig 4D).

Intracellular microRNA levels have also been shown to be regulated by active export or secretion of miRs, including from hepatocytes/liver (Bala et al, 2012; Momen-Heravi et al, 2015; Erhartova et al, 2019; Lee & Kim, 2022; Pozniak et al, 2022). Hence, we wanted to check if hepatic fed miR oscillations were causally or consequentially associated with abundance of these miRs in circulation. Not surprisingly, we could not detect pri- and pre-miRs in the serum, consistent with very few studies that have reported detectable levels of circulatory pri- and pre-miRs. Nonetheless, we observed an oscillatory pattern in the levels of mature miRs in serum under ad libitum–fed, 24-h–starved, and 6-h–refed conditions, as indicated (Fig 4E). It was interesting to note that the abundance of mature miRs mirrored the changes we observed in hepatocytes/liver (Figs 1A and 4E). Together, these results clearly illustrated that the fed-fast cycle–dependent changes in hepatic miRs, described here, are largely dictated by anticipatory biogenesis and maturation within hepatocytes and possibly independent of secretion.

## MiR biogenesis is perturbed in aged and over-nutrition mice liver

Others and we have earlier demonstrated the importance of transcriptional and post-transcriptional mechanisms in maintaining physiological homeostasis and causal effects of aberrant fed-fast cycles in aging and metabolic deficits (Suliga et al, 2015; Maniyadath et al, 2019; Chattopadhyay et al, 2020; Bideyan et al, 2021). In this regard, we profiled pri- and pre-miRs in separate paradigms of over-nutrition and aging. It is interesting to note that we observed a dramatic reduction in the starvation-dependent up-regulation in miR biogenesis in aged mice when compared to the young mice (Fig 5A and B). Specifically, whereas starvation-dependent induction of primary transcripts of let-7i and miR-204 was significantly dampened (Fig 5A and B), refed-mediated

reduction in pri-miR-221/-222 was lost in aged mice (Fig 5A). Importantly, fed-fast-refed oscillation in pre-miRs was conspicuously subdued in livers isolated from aged mice (Fig 5A and B).

To study the effect of over-nutrition, we assayed for pri-/pre-miRs in livers harvested from mice fed with high-fat diet (Fig 5C and D) and 10% sucrose in water (along with normal chow diet) (Fig 5E and F). We found diminished starvation response, in pri-/pre-miRs, in both of these models when compared to chow diet–fed mice (Fig 5C–F). Overall, these results suggest that the anticipatory up-regulation of miR biogenesis is lost/dampened under contexts which are associated with perturbed physiological homeostasis in chronic over-nutrition paradigms.

## Starvation signals regulate miR biogenesis via PPARα in hepatocytes

Having demonstrated starvation dependence, we wanted to further investigate the molecular factors that mediate the up-regulation of pri-/pre-miRs in hepatocytes. The hepatic starvation response is collectively governed by autonomous metabolic signalling/sensing and non-autonomous inputs predominantly from glucagon. In this regard, we assayed for primary and precursor miR transcripts in primary hepatocytes grown in medium containing differential glucose concentrations, viz., 0 mM (no glucose—NG), 5 mM (low glucose—LG), and 25 mM (high glucose—HG), to mimic fasting/fed states (Fig S5A).

It was interesting to find that whereas pri-/pre-miR-204 showed a robust up-regulation in LG and NG, there was a mild induction in pre-miR-222, which indicated a partial glucose-deprivation–dependent control (Fig S5A). Furthermore, treating hepatocytes with glucagon, to evaluate the dependence on starvation endocrine inputs, resulted in elevated levels of pri-/pre-miRs (Fig 6A). To further substantiate these, we used forskolin, a known activator of adenylate cyclase, which is downstream of the glucagon receptor, and observed a consistent induction that was akin to the response seen during starvation (Fig 6B). To pharmacologically mimic a starvation state that may also indicate the feasibility of therapeutic intervention, we treated primary hepatocytes with metformin, which is an FDA-approved clinically administered drug to induce catabolic state. Interestingly, metformin treatment led to an increase in miR biogenesis (Fig 6C).

Next, we wanted to check if starvation-sensitive transcription factors are involved in the miR-biogenic program that we have described so far. Among others, PPARα is key for mediating the transcription of several hepatic genes in response to fasting. Hence, we treated primary hepatocytes with WY-14643, a specific agonist of

**Figure 2. Temporally restricted feeding paradigms reprogram the oscillatory miRNA biogenesis in the liver.**
**(A)** Schematic outline of the feeding paradigm used. Dark-fed and light-fed mice were allowed access to food from ZT12-ZT24 and ZT0-ZT12, respectively. **(B)** Respiratory exchange ratio of dark-fed and light-fed mice over a 24-h light-dark cycle (n = 3). **(C)** Relative expression of *Cry1* mRNA from the liver of dark-fed and light-fed mice at the indicated ZT. *Cry1* transcript level was normalized to *Actin* mRNA level and plotted as fold change with respect to ZT4 for both dark-fed and light-fed mice (N = 2, n = 3). **(D)** Heatmap depicting relative expression of the starvation-responsive gene mRNAs at the indicated ZT. mRNA levels of the respective genes were normalized to *Actin* mRNA level and plotted as fold change with respect to ZT4 for both dark-fed and light-fed mice (N = 2, n = 3). **(E, F)** Relative expression of (E) Pri-miRs and (F) Pre-miRs from the liver of dark-fed and light-fed mice at the indicated ZT. Pri- and pre-miR-transcript levels were normalized *Actin* mRNA level and plotted as fold change with respect to ZT4 for both dark-fed and light-fed mice (N = 2, n = 3). Data information: In (B, C, E, F), data are represented as mean ± SEM. For (C, E, F), statistical significance between dark-fed and light-fed groups at indicated ZT was calculated using multiple *t* tests with Holm-Sidak correction. For (D), statistical significance was calculated using one-way ANOVA with Tukey's test for multiple comparisons at indicated ZT with respect to ZT4 (*P < 0.05; **P < 0.01; ***P < 0.001).

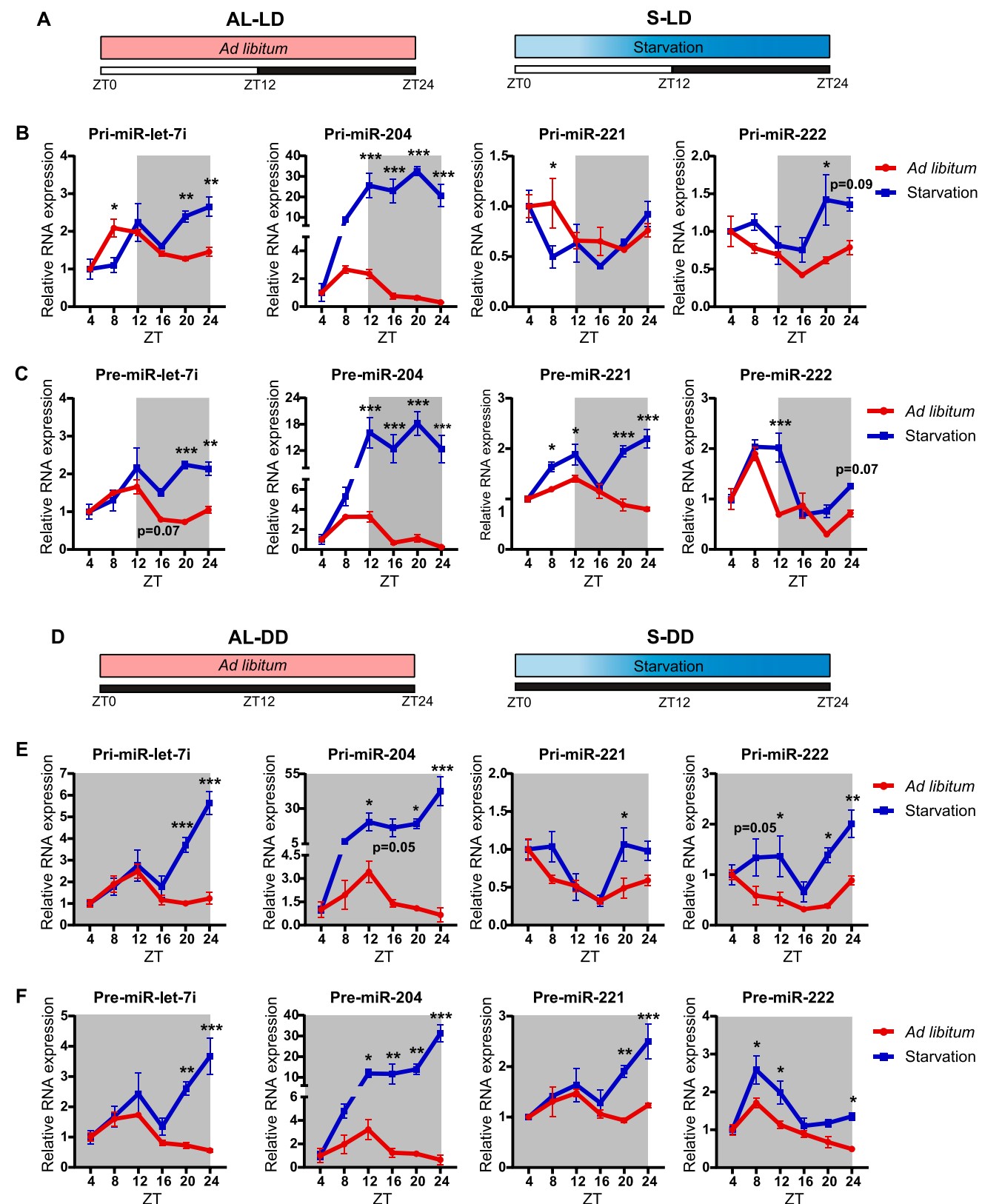

**Figure 3. Hepatic miRNA biogenesis is responsive to progressive starvation and is independent of light inputs.**
**(A)** Schematic outline of the feeding paradigms used. Mice were kept under a normal 12-h light-dark cycle and either provided ad libitum access to feed (AL-LD) or starved from ZT0 (S-LD) for different durations (4, 8, 12, 16, 20, and 24 h), respectively. **(B, C)** Relative expression of (B) Pri-miRs and (C) Pre-miRs from the liver of AL-LD and

PPARα, and found that the pri- and pre-levels were up-regulated at both 10 h (Fig S5B) and 24 h (Fig 6D). Moreover, administration of bezafibrate, which is used in clinics, also resulted in increased expression, phenocopying the effects observed with WY-14643 (Fig S5C). Together, these findings clearly indicate that anticipatory expression of hepatic fed miRs is associated with activation of metabo-endocrine factors, akin to starvation, and importantly hint at the potential of using pharmacological agents to restore miR homeostasis.

## Discussion

Efficient physiological toggling between anabolic and catabolic pathways during fed-fast cycles is essential for metabolic homeostasis, and an inability to do so has been associated with aging and non-communicable diseases (Ribarič, 2012; Geisler et al, 2016; Petersen et al, 2017; Paoli et al, 2019; Chattopadhyay et al, 2020). Given that refeeding following starvation is often unanticipated, it is intuitive to expect that robust mechanisms would allow organisms to rapidly transit from fasted to a refed state (Jitrapakdee, 2012; Bideyan et al, 2021). Whereas much is known about cephalic mechanisms (Power & Schulkin, 2008), molecular factors that govern gene expression programs and create anticipatory regulatory loops remain largely unknown. Furthermore, owing to the fact that feeding and fasting are intricately coupled with light-dark or circadian cycles (Gerhart-Hines & Lazar, 2015; Reinke & Asher, 2019; Pickel & Sung, 2020), it is important to delineate the roles of these zeitgebers in dictating molecular anticipation. In this context, we have unravelled the significance of metabolic and endocrine inputs in exerting mechanistic control over molecular processes that govern the anticipatory biogenesis of hepatic fed microRNAs. Importantly, we demonstrate the differential contribution of fed-fast and circadian cycles in controlling miR homeostasis in hepatocytes.

Post-transcriptional control of gene expression is known to provide energy-/time-efficient and dynamic spatio-temporal tuning of cellular functions (Halbeisen et al, 2008; Hocine et al, 2010). This is particularly relevant for fed-fast cycles because rapid and reversible inhibition of translation is essential (Koike et al, 2012; Le Martelot et al, 2012). Consistent with this, others and we have demonstrated the importance of microRNAs in governing hepatic physiology (Hu et al, 2012; Du et al, 2014; Maniyadath et al, 2019). Strikingly, we also found that oscillatory-fed microRNAs, which inhibit the expression of catabolic genes and consequently enable fasted-refed transition, are expressed in anticipation during a starvation state (Maniyadath et al, 2019). Albeit parallel studies have also illustrated circadian dependence or rhythmicity in the expression of microRNAs across tissues, including the liver (Menet et al, 2012; Vollmers et al, 2012; Ji et al, 2023), whether fed-fast cycles

contribute to miR biogenesis independently or cooperatively (with circadian inputs) remains unknown. In this context, we have delineated the causal upstream cues that dictate the anticipatory expression of hepatic fed microRNAs, especially for let-7i, mir-204, mir-221, and mir-222, which not only displayed the most robust oscillation but also otherwise have been shown to be important for liver functions and hepatic cancers (Chen et al, 2014; Luo et al, 2017; Song et al, 2017). Our findings also provide confirmatory evidence for transcription-driven anticipatory expression during starvation. Using classical paradigms to perturb circadian and fed-fast cycles, we establish that whereas let-7i and mir-204 are tuned solely by fed-fast cues, mir-221 and mir-222 show only partial dependence. Specifically, comparing pri-miR levels in S-LD and S-DD showed that prolonged starvation induced heightened expression of let-7i and miR-204, to similar extents, clearly indicating that they are not dependent on circadian inputs. This also corroborated starvation dependence from AL-LD and S-LD paradigms.

miR processing has been shown to be regulated, and several reports have indicated differential Drosha/DGCR8 and Dicer activities to affect pri-, pre-, and mature-miR levels (Creugny et al, 2018; Vergani-Junior et al, 2021). Even though post-translational modifications, co-factors, RNA structures, and relative abundances of miR-transcripts have been proposed to impinge on Drosha/DGCR8 and Dicer-dependent maturation of pri-miRs to pre-miRs and mature miRs, respectively (Heo et al, 2008; Davis & Hata, 2009; Feng et al, 2011; Conrad et al, 2014; Alarcón et al, 2015; Louloupi et al, 2017), if/how this leads to global versus miR-specific effects remains less understood. This is relevant because we observed that, whereas pre-miR-221/222 exhibited similar expression profile in ad libitum condition, they showed mutually exclusive expression profiles in response to continuous starvation. This differential response was conspicuous in both the starvation paradigms (S-LD and S-DD) and although intriguing, did suggest an intricate regulatory mechanism that determines miR homeostasis, whose mechanistic basis will have to be unravelled in the future. An equally intriguing finding was fasted-refed transition dependent processing of pre-miRs to mature miRs, which is possibly mediated by anabolic inputs as an additional layer of regulation. Consistent with this, our results show that insulin signalling impinges on the maturation or processing of pri-/pre-miRs. Given that ERK, which is downstream of insulin signalling, has been otherwise shown to affect maturation of let-7 miRNA (Heo et al, 2008; Liu et al, 2017), we posit that upstream metabolic and neuroendocrine inputs associated with a refed state are required for hepatic miR maturation. We further propose an interplay between metabolism and DICER expression, which fluctuates during fed-fast-refed transitions, in eliciting oscillatory patterns for mature-miR abundance in the liver.

Another key highlight of the study is the observation of deregulated expression of hepatic fed miRs in response to dietary

---

S-LD mice, at the indicated ZT. Pri- and pre-miR-transcript levels were normalized to *Actin* mRNA level and plotted as fold change with respect to ZT4 (N = 2, n = 3). **(D)** Schematic outline of the feeding paradigms used. Mice were kept under constant darkness for 24 h and either provided ad libitum access to feed (AL-DD) or starved from ZT0 (S-DD) for different durations (4, 8, 12, 16, 20, and 24 h). **(E, F)** Relative expression of (E) Pri-miRs and (F) Pre-miRs from the liver of AL-DD and S-DD mice, at the indicated ZT. Pri- and pre-miR-transcript levels were normalized to *Actin* mRNA level and plotted as fold change with respect to ZT4 (N = 2, n = 3). Data information: AL-LD—ad libitum light-dark, S-LD—starvation light-dark, AL-DD—ad libitum dark-dark, S-DD—starvation dark-dark; in (B, C, E, F), data are represented as mean ± SEM. Statistical significance between ad libitum–fed and progressively starved mice at indicated ZT was calculated using multiple *t* tests with Holm-Sidak correction (*$P < 0.05$; **$P < 0.01$; ***$P < 0.001$).

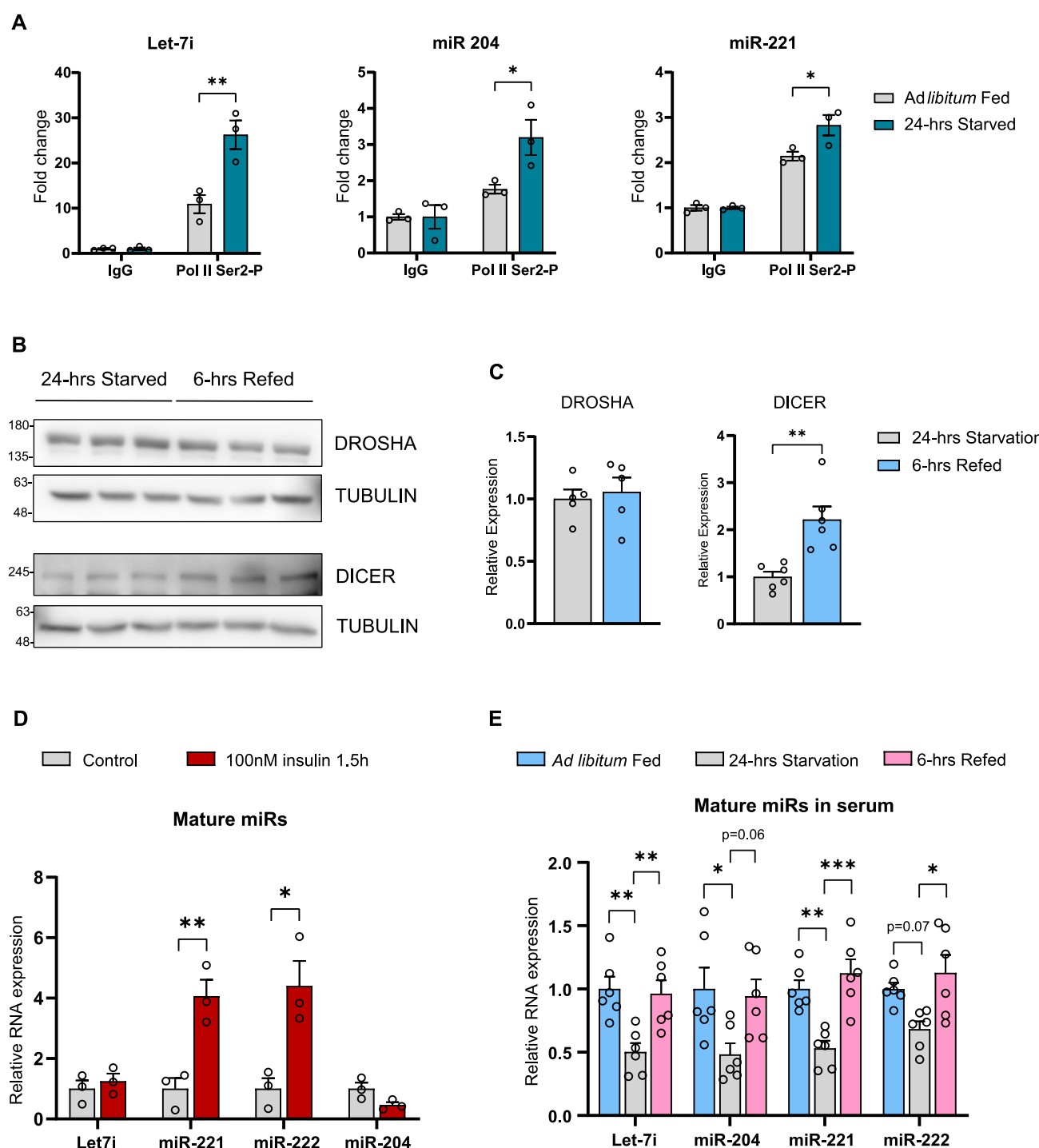

**Figure 4. Transcriptional and post-transcriptional processing control biogenesis of hepatic fed miRs.**
**(A)** ChIP-qRT-PCR of S2P-Pol II binding at the microRNA genomic loci in ad libitum–fed and 24-h–starved mice liver represented as fold change with respect to IgG.
**(B, C)** Protein expression of DROSHA and DICER in 24-h–starved and 6-h–refed mice liver (B) Representative immunoblots and (C) quantification of DROSHA and DICER band intensities. DROSHA/DICER band intensities were quantified with tubulin as a loading control and presented as fold change in comparison to ad libitum fed (N = 2, n = 2–3). **(D)** Relative expression of mature miRs in primary hepatocytes treated with 100 nM insulin for 1.5 h. Mature miR-transcript levels were normalized to 18S rRNA level and plotted as fold change with respect to control (N = 2, n = 3). **(E)** Relative expression of mature miRs in serum isolated from ad libitum fed, 24-h–starved, and 6-h–refed mice. Mature miR-transcript levels were normalized to Cel-miR-39-3p spike-in control level and plotted as fold change with respect to it (N = 2, n = 3). Data information: In (A, C, D, E), data are represented as mean ± SEM. For (A), statistical significance between groups was calculated using two-way ANOVA with Tukey's test for multiple comparisons. For (C, D), statistical significance was calculated using a $t$ test. For (E), statistical significance was calculated using multiple $t$ tests with Holm-Sidak correction (*$P < 0.05$; **$P < 0.01$; ***$P < 0.001$).

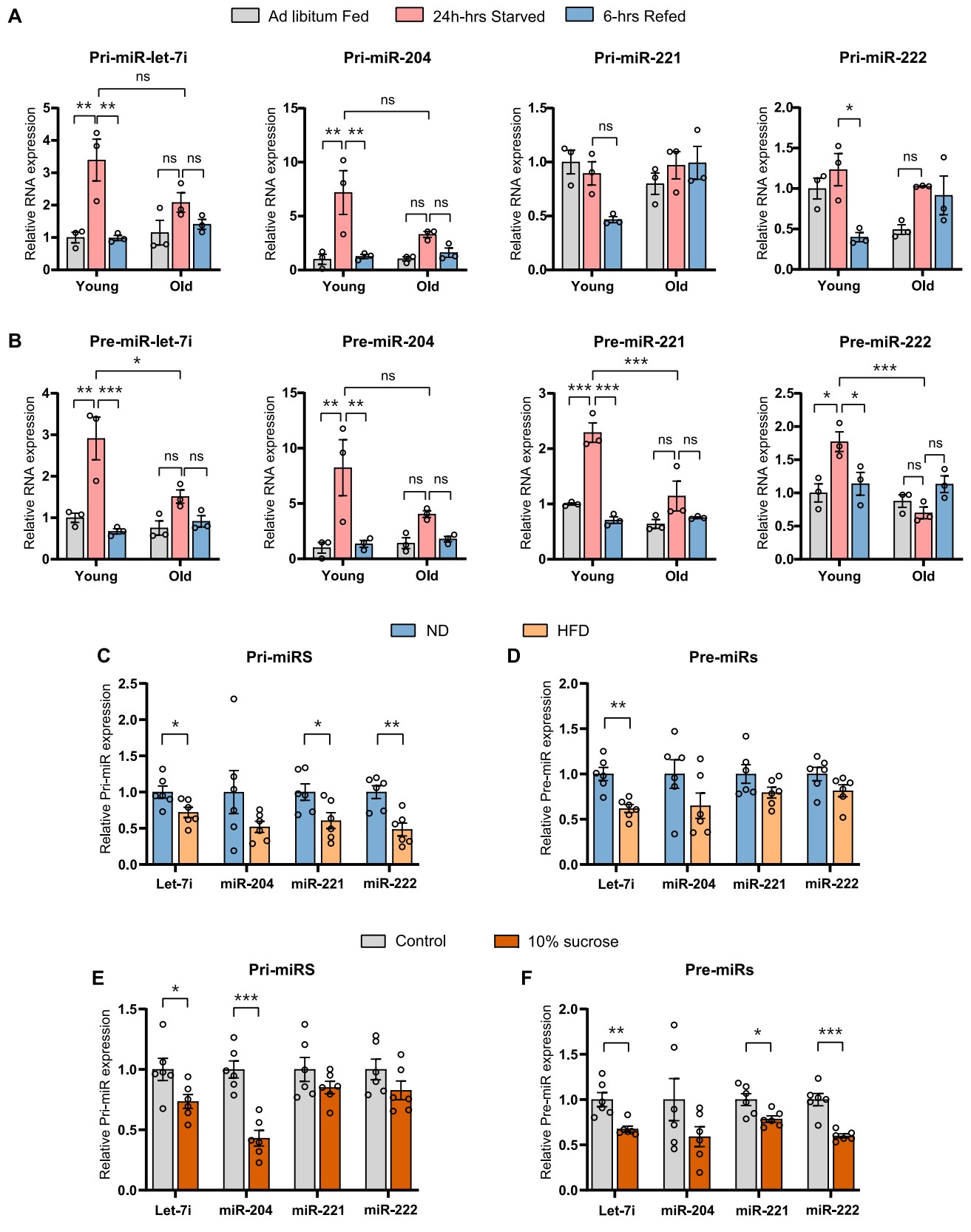

perturbations and aging. Whereas qualitative and quantitative changes in miRs, intracellular and circulating, during aging and pathological conditions are well documented (Noren Hooten et al, 2013; Jung & Suh, 2014; Deiuliis, 2016), whether these are resultants of altered miR biogenesis needs to be unravelled. In this context, we have observed that the biogenesis of microRNAs profiled in this study is significantly dampened in aging and over-nutrition paradigms. Notably, feeding mice with high-fat diet (HFD) and sucrose water (along with a standard chow diet) muted the biogenesis of let-7i, miR-204, miR-221, and miR-222. Together, these clearly suggest that both short-term and long-term dietary/metabolic inputs play a significant role in orchestrating miR homeostasis in the liver. It will be interesting to investigate, in the future, if such perturbations affect miR biogenesis, degradation, and exosome-mediated secretion, globally.

Given the complex interplay between metabolism, miR homeostasis, and organismal physiology, which are in turn dependent upon fed-fast and circadian cycles (Rottiers & Näär, 2012; Dumortier et al, 2013; Hartig et al, 2015), it is nearly impossible to establish whether miR biogenesis acts as an initiator or mediator mechanism. Nonetheless, dissecting molecular factors and evaluating the modulatory impact of commonly used therapeutic interventions becomes crucial. In this regard, we not only provide mechanistic insights into key factors that link metabolic inputs to miR biogenesis but also illustrate that pharmacological agents which regulate nodal molecular factors, impinge on hepatic fed miR homeostasis. We clearly illustrate PPARα as one of the starvation-dependent factors that exert control over the anticipatory expression of miRs. Importantly, treating hepatocytes with metformin and bezafibrate, which are well established to activate AMPK and PPARα-dependent metabolic rewiring (Zhou et al, 2001; Puligheddu et al, 2013), led to a robust response vis-à-vis miR biogenesis. Furthermore, taken together with our earlier report, the findings presented here raise the possibility of using FDA-approved drugs to modulate miR expression to help mitigate physiological deficits associated with metabolic diseases.

In conclusion, our study unequivocally delineates fed-fast and circadian-dependent expression of hepatic fed miRs, which is pivotal for molecular anticipation. Using paradigms that recapitulate normal physiological settings, we show a complex interplay between metabolism and miR biogenesis in a miR-dependent manner. Owing to the detrimental impact of modern lifestyles, largely mediated by perturbed fed-fast and circadian cycles (Arble et al, 2010; Charlot et al, 2021), we have unravelled physiological inputs that choreograph oscillatory/anticipatory microRNA biogenesis in the liver. Even though miRs have emerged as clinically feasible therapeutic targets, current approaches are based on miR-mimics and anti-miRs that affect individual microRNAs (Baumann &

Winkler, 2014; Chakraborty et al, 2021; Diener et al, 2022). Whereas this has merits, miRs are known to modulate cellular/organismal functions by exerting additive and convergent control over the expression of target mRNAs. Therefore, identifying physiological contexts and/or pharmacological interventions that reset miR homeostasis, possibly by regulating a network of miRs as demonstrated in this study, will likely yield higher dividends.

# Materials and Methods

## Animal experiments

Animal experiments were performed using C57BL6NCrl mice that were reared under standard animal house conditions (12-h light-dark cycle and fed a regular chow diet). 2–3-mo-old male mice were used for all diet and circadian perturbations, unless specified otherwise. For aged cohorts, 20–22-mo-old male mice were used. The procedures and the project were approved and were in accordance with the Tata Institute of Fundamental Research Institutional Animal Ethics Committee (CPCSEA-56/1999).

Fed-starved-refed paradigms: (a) Fed mice—ad libitum–fed mice euthanized at ZT1 (8.00 AM) (b) Starved mice—ad libitum–fed mice starved for 24 h and euthanized at ZT1 (8.00 AM) (c) Refed mice—post 24 h of starvation, mice were fed ad libitum for 6 h and euthanized at ZT7 (2.00 PM). Both young (2–3 mo old) and aged (20–22 mo old) male mice were used for these experiments, as indicated.

Light-fed and dark-fed paradigms: mice were entrained for 14 d for time-restricted feeding with exclusive access to food either during the light phase (ZT0–ZT12) or the dark phase (ZT12–ZT24), as indicated.

Light-dark ad libitum (LD-AL) and starvation (LD-S) paradigms: mice were subjected to starvation by removing the feed at ZT0 (7 AM), designated as LD-S. Mice that were ad libitum fed served as controls, designated at LD-AL. Both LD-S and LD-AL mice were euthanized at the same ZT time points at 4-h intervals (starting from ZT4).

Dark-dark paradigms: mice housed under constant darkness for 24 h were subjected to different durations of starvation by removing feed at ZT0, designated as DD-S. DD-S mice were compared with those that had ad libitum access to feed during the continuous dark phase, designated as DD-AL. In both cases, animals were euthanized at 4-h intervals (as indicated), and all mouse handling and tissue harvesting were performed under dim red light.

HFD and sucrose supplementation paradigms: (a) mice were provided ad libitum access to normal chow or 35% HFD for 12 wk and (b) normal chow diet ad libitum–fed mice were given access to only water or 10% sucrose in water for 12 wk. For both paradigms, mice

**Figure 5. miRNA biogenesis is perturbed in aged and over-nutrition mice livers.**
**(A, B)** Relative expression of (A) Pri-miR and (B) Pre-miR in ad libitum–fed, 24-h–starved, and 6-h–refed young (2–4 mo old) and old (20–22 mo old) mice livers. Pri- and pre-miR-transcript levels were normalized to *Actin* mRNA level and plotted as fold change with respect to ad libitum fed (N = 2, n = 3). **(C, D)** Relative expression of pri-(C) and pre-miRs (D) in 12 h starved mice livers subjected to 12 wk of 35% high-fat diet. **(E, F)** Relative expression of pri-(E) and pre-miRs (F) in 12-h–starved mice livers subjected to 12 wk of 10% sucrose water (with regular chow). Pri- and pre-miR-transcript levels were normalized to *Actin* mRNA level and plotted as fold change with respect to (C, D) control diet and (E, F) normal water (with regular chow) (N = 2, n = 3). Data information: In (A, B, C, D, E, F), data are represented as mean ± SEM. For (A, B), statistical significance between groups was calculated using two-way ANOVA with Tukey's test for multiple comparisons. For (C, F), statistical significance was calculated using a *t* test (*$P < 0.05$; **$P < 0.01$; ***$P < 0.001$).

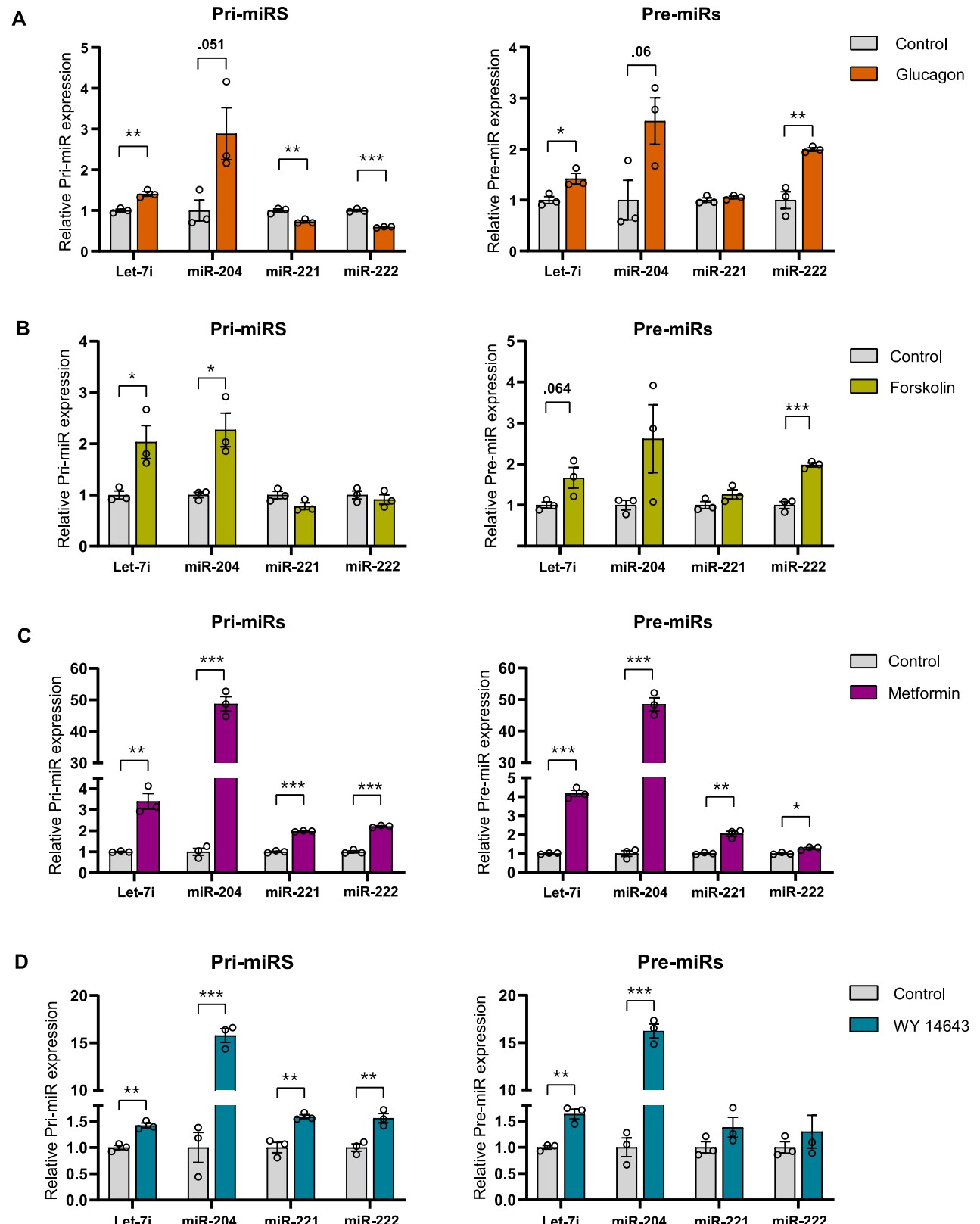

**Figure 6. Starvation cues regulate miR biogenesis in primary hepatocytes.**
**(A, B, C, D)** Relative expression of pri-miRs and pre-miR in primary hepatocytes treated with 30 nM of glucagon in LG media for 6 h (A), 10 μM of forskolin in LG media for 12 h (B), 2 mM of metformin in HG media for 12 h (C), 50 μM of WY-14643 in HG media for 24 h (D). Pri- and pre-miR-transcript levels were normalized to *Actin* mRNA level and

were starved for 12 h starting at ZT13 (8 PM) and euthanized at ZT1 (8 AM).

### RER measurements

For RER measurement, $O_2$ consumption and $CO_2$ production were measured using indirect calorimetry in an Oxymax Comprehensive Lab Animal Monitoring System CLAMS open circuit system (Columbus Instruments). Animals were housed in individual metabolic cages with a constant airflow of 0.5 liters/min and under standard conditions. Readings from each cage were measured every 20 min for 48 h (12 h light/12 h dark cycle). The RER was derived from the measured volumes of oxygen consumed ($VO_2$) and carbon dioxide produced ($VCO_2$) that were normalized to body mass (ml/kg/h).

### Primary hepatocyte isolation and culture

10–12-wk-old male mice were used for primary hepatocyte isolation as described earlier (Maniyadath et al, 2019). Briefly, the liver was perfused with Buffer A (HBSS, 1 mg/ml D-glucose, 25 mM HEPES, and 0.5 mM EGTA) and digestion medium low glucose DMEM (LG, 5 mM glucose), antibiotic antimycotic solution (AA), 15 mM HEPES, and 19 mg collagenase (type IV) in a sequential manner. The perfused livers were then harvested and minced in the digestion media. It was further passed through a 70-micron pore-size filter and centrifuged at 50$g$ for 5 min at 4°C to obtain the hepatocyte pellet.

The hepatocyte pellet was washed three times with high glucose DMEM media (HG, 25 mM glucose) and plated on cell culture plates/dishes coated with collagen and in a medium containing 10% FBS, as described earlier (Maniyadath et al, 2019). Six hours post plating, the cells were cultured overnight in serum-free HG media. Hepatocytes were cultured in high glucose (HG: 25 mM), low glucose (LG: 5 mM) or glucose-free media (NG) and all treatments were performed in either of these conditions, as indicated: (a) 2 mM metformin in HG media for 12 h; (b) 30 nM glucagon for 6 h and 10 $\mu$M forskolin for 12 h in LG media; (c) PPAR$\alpha$ agonists 50 $\mu$M WY-14643 and 100 $\mu$M bezafibrate for 10 and/or 24 h in HG media; (d) 100 nM of insulin for 1.5 h in HG media. Following these treatments, cells were collected in TriZol and proceeded for RNA isolation.

### RNA isolation, cDNA synthesis, and quantitative PCR (qRT-PCR)

RNA isolation, cDNA synthesis, and qRT-PCR were carried out as per the manufacturer's instructions. Briefly, total RNA was isolated from liver tissue/hepatocytes using TriZol reagent, and 5–8 $\mu$g RNA was used to prepare cDNA using random hexamers and SuperScript-IV kit. qRT-PCR was performed using KAPA SYBR FAST Universal 2X qRT-PCR Master Mix on Roche Light Cycler 480 II and LC 96 instruments. For mature microRNA profiling, the cDNA was prepared using the QuantiMir RT Kit, and qRT-PCR was performed as per the manufacturer's instructions. For normalization, levels of actin mRNA/18S rRNA/Cel-miR-39-3p spike-in control were used. Primer sequences are given in Table S1.

### Serum microRNA isolation

Blood samples of mice subjected to ad libitum–fed, 24-h–starved, and 6-h–refed conditions were collected through retro-orbital bleeding. These blood samples were further subjected to centrifugation at 3,000$g$ for 15 min at 4°C, and the resulting serum was then stored at –80°C for subsequent use. Serum RNA isolation was performed using the miRNeasy Serum/Plasma Kit as per the manufacturer's instructions. Briefly, total RNA from 100–200 $\mu$l of mouse serum was isolated using the QiaZol Reagent. An equal amount of Cel-miR-39-3p spike-in control was added to each serum sample at the onset of the protocol.

### Chromatin immunoprecipitation

Mice kept under ad libitum–fed and 24-h–starved conditions were euthanized and livers were harvested, chopped into pieces, and washed in cold PBS. The tissue was fixed in 1% formaldehyde for 10 min at RT (with continuous rocking) and subsequently treated with 125 mM of glycine for 5 min followed by two PBS washes.

For the preparation of chromatin, around 100 mg of fixed tissue was homogenized in cell lysis buffer (10 mM Tris–HCl pH 8.0, 10 mM NaCl, NP40), lysed for 10 min on ice, and centrifuged at 1,300$g$ for 5 min. The nuclear pellet obtained was further washed twice with cell lysis buffer, followed by lysis in nuclear lysis buffer (50 mM Tris–HCl pH 8.0, 10 mM EDTA, 1% SDS, 1% NP40) for 10 min on ice. The lysates were diluted using ChIP dilution buffer (CDB, 0.01% SDS, 1.2 mM EDTA, 16.7 mM Tris–HCl pH 8.0, 167 mM NaCl), snap-frozen in liquid $N_2$, and thawed on ice. Thawed lysates were sonicated in Bioruptor Pico sonicator (Diagenode) using 30 s ON/30 s OFF for 5–10 cycles. The sonicated lysates were centrifuged at 15,000$g$ for 10 min, and the supernatant containing sheared chromatin was used first to check proper shearing followed by immunoprecipitation.

Magnetic Protein A beads blocked in CDB containing 0.01% of BSA and 0.1 mg/ml of yeast tRNA were used for pull down. Chromatin lysates were diluted with CDB and then precleared for 1–2 h at 4°C. Precleared lysates were incubated with S2P-Pol II/IgG antibodies overnight at 4°C. 5% (V/V) of the precleared chromatin was stored as input control. The antibody-protein complexes were pulled down by blocked Protein A beads. After immunoprecipitation, beads were washed with cold buffers, in the following sequence: 2X Low Salt Buffer (0.1% SDS, 1% Triton X-100, 2 mM EDTA, 20 mM Tris–HCl pH 8.0, 150 mM NaCl), 2X High Salt Buffer (0.1% SDS, 1% Triton X-100, 2 mM EDTA, 20 mM Tris–HCl pH 8.0, 500 mM NaCl), 2X Lithium Chloride Buffer (0.25 M LiCl, 1% NP40, 1% sodium deoxycholate, 1 mM EDTA, 10 mM Tris–HCl pH 8.0), 2X Tris-EDTA Buffer (10 mM Tris–HCl, 1 mM EDTA) and eluted for 30 min at 65°C in Elution Buffer (EB, 1% SDS, 0.1 M NaHCO3). Eluates were incubated with NaCl (conc. 200 mM) and sequentially treated with ribonuclease A (conc. 1 $\mu$g/$\mu$l) and proteinase K (at conc. 250 $\mu$g/ml). Final DNA was isolated using the standard Phenol: Chloroform DNA isolation method, and extracted DNA samples were resuspended in nuclease-free

plotted as fold change with respect to control (N = 2, n = 3) (A, B, C, D). Data information: LG—low glucose, HG—high glucose. In (A, B, C, D), data are represented as mean ± SEM, and statistical significance was calculated using a $t$ test (*$P$ < 0.05; **$P$ < 0.01; ***$P$ < 0.001).

distilled water. Enrichment of S2P-Pol II was examined using qRT-PCR (primer sequences are given in Table S1).

## Western blotting

An equal amount of protein (50–100 μg) samples was run on SDS–PAGE and transferred to PVDF membrane (Millipore). The membranes were then blocked in 5% skimmed milk in 0.1% TBS-Tween 20 at RT for 1 h on the rocker. Blots were cut according to molecular weights indicated by a pre-stained ladder (Abcam), and appropriate primary antibodies were added and incubated overnight (12–16 h) at 4°C. Blots were washed and incubated with appropriate secondary antibodies. After the secondary antibody incubation, the blots were washed thrice, and the Supersignal West Pico PLUS or Supersignal West Femto ECL kit (Thermo Fisher Scientific) was used to detect the bands on GE Amersham Imager 600. All antibodies, chemicals, and reagents used in this study are given in Table S2.

## Quantification and statistical analysis

Data are expressed as means ± SEM. Statistical analyses were performed using GraphPad Prism (version 8.0). $t$ test and ANOVA were used to determine statistical significance. A value of $P \leq 0.05$ was considered statistically significant. $*P \leq 0.05$; $**P \leq 0.01$; $***P \leq 0.001$.

# Supplementary Information

# Acknowledgements

This research has been supported by the following funding sources: TIFR/DAE (19P0116), TIFR/DAE (19P0911), and Department of Biotechnology (BT/PR29878/PFN/20/1431/2018) to U Kolthur-Seetharam. We thank ACTREC Mumbai for providing us with C57BL/6NCrl mice. We thank the Advanced Research Unit on Metabolism, Development, and Aging (ARUMDA) Consortium for supporting a part of this study. We especially thank Dr. Kalidas Kohale, Dr. Shital Suryavanshi, Chetan Sable, TIFR Mumbai, and IISER Pune animal house staff for their help with the animal experiments. We are thankful to Dr. Mahesh J Kulkarni from NCL-Pune for providing us, Metformin-HCl as a kind gift. We are grateful to Dr. Jomon Joseph from NCCS, Pune, for sharing the DICER and DROSHA antibodies with us. We extend our acknowledgement to UK laboratory members for their critical input and discussions during the study.

## Author Contributions

S Usha Satheesan: conceptualization, data curation, formal analysis, validation, investigation, visualization, methodology, and writing—original draft, review, and editing.
S Chowdhury: data curation, formal analysis, validation, investigation, visualization, methodology, and writing—original draft, review, and editing.

U Kolthur-Seetharam: conceptualization, resources, data curation, formal analysis, supervision, funding acquisition, validation, visualization, methodology, project administration, and writing—original draft, review, and editing.

## Conflict of Interest Statement

The authors declare that they have no conflict of interest.

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
