## [Reviewer comments · Life Science Alliance]

Life Science Alliance

Metabolic and Circadian Inputs Encode Anticipatory Biogenesis of Hepatic Fed MicroRNAs

U. S. Sandra, Shreyam Chowdhury, and Ullas Kolthur-Seetharam

DOI: <https://doi.org/10.26508/lsa.202302180>

Corresponding author(s): *Ullas Kolthur-Seetharam, Tata Institute of Fundamental Research*

Review Timeline:

Submission Date:	2023-05-24
Editorial Decision:	2023-07-24
Revision Received:	2024-01-14
Editorial Decision:	2024-02-06
Revision Received:	2024-02-13
Accepted:	2024-02-13

Scientific Editor: *Eric Sawey, PhD*

Transaction Report:

July 24, 2023

Re: Life Science Alliance manuscript #LSA-2023-02180-T

Prof. Ullas Kolthur-Seetharam
Tata Institute of Fundamental Research
Dept of Biological Sciences
Homi Bhabha Road, Colaba
Mumbai, Maharashtra 400005
India

Dear Dr. Kolthur-Seetharam,

Thank you for submitting your manuscript entitled "Metabolic and Circadian Inputs Encode Anticipatory Biogenesis of Hepatic Fed MicroRNAs" to Life Science Alliance. The manuscript was assessed by expert reviewers, whose comments are appended to this letter. We invite you to submit a revised manuscript addressing the Reviewer comments.

Thank you for this interesting contribution to Life Science Alliance. We are looking forward to receiving your revised manuscript.

Sincerely,

B. MANUSCRIPT ORGANIZATION AND FORMATTING:

Reviewer #1 (Comments to the Authors (Required)):

In the submitted manuscript titled 'Metabolic and Circadian Inputs Encode Anticipatory Biogenesis of Hepatic Fed MicroRNAs' by Sandra et al, the authors have followed the initial finding reported earlier by the same group (Maniyadath et al., 2019) to identify the oscillatory expression of certain miRNAs that are known to be controller of important starvation responsive genes. The investigation reported here, they have concluded an interesting hypothesis explaining the observation. They have scored the anticipatory high expression of pri- and pre-miRNA of corresponding mature miRNAs that are shown to be down regulated in the process of starvation. This hypothesis has been tested in the context of day light cycle and circadian rhythm and in the context of hypoglycemic and hyperglycemic conditions induced by pharmacological agents. I appreciate the approaches the authors have taken but I am disappointed of not finding any "real" mechanistic insight at molecular level that can explain these observations.

The key concerns I have:

1. How the author can be certain that the upregulation (anticipatory) of pr- and pre-miRNAs are happening at post-transcription level and not due to a processing defect caused by the lowering of some components of miRNA machineries that are known to be regulated (like Dicer, or DGCR8) under different metabolic context. The expression of these components should be checked.
2. To confirm the regulation happening at transcriptional level, the authors must have done experiments by doing DNA CHIP assay to confirm the enhanced association of the polymerase with the promoter region of the responsive miRNA expressing locus in starvation . It is also important to have control with the miRNA gene locus that are not expressing in the same manner during starvation-fed cycle.
3. The extracellular export of miRNA can also contribute to the level of changed miRNA. It will be wise to look at the circulatory level of target miRNAs, pre-and pri-miRNAs levels in the serum of the animals going through fed and starvation cycle.
4. The anticipatory differential expression of mature and precursor levels of miRNAs at the global level should be wisely followed in an bioinformatic approach involving gene expression data from normal fed and starved animal liver. This is essential to find a true link related to metabolic dysregulation and conciliatory miRNA expression-the basis of the hypothesis.

Reviewer #2 (Comments to the Authors (Required)):

Anticipatory behavior and physiological response to food is particularly important for animals to restore metabolic homeostasis promptly and increase the survival/fitness. miRNAs have been identified as important posttranscriptional regulators of gene expressions including hepatic metabolism. In this manuscript, the authors report several hepatic miRNAs that respond to fed at the biogenesis level.

Overall this is an interesting paper mainly focusing on the biogenesis of miRNAs responding to metabolic stages. The findings are interesting and the current evidence seem solid. However, I do have a few things that need to be addressed.

1. In the abstract, the authors mentioned that "we report anticipatory biogenesis of oscillatory hepatic microRNAs that peak during a fed state and inhibit starvation responsive genes". This might have been reported by the authors and others in previous papers, however, I don't see any "inhibit starvation responsive gene"expression data throughout the current manuscript. Indeed, this is also one of the major caveats for this manuscript. The authors systematically checked the biogenesis of the four miRNAs under different conditions, however there is almost no data of their effects on potential targets. I suggest the authors used their samples to run a few qPCR or even WB to check several potential miRNA targets that might be starvation related here.
2. This manuscript lacks explanations of the rationale to study these four oscillatory miRNAs in the beginning of their results. Is this from a screen? Why do they start from these four? If just because they identified these four miRNAs in the previous study, then why they want to do it again? What new does this study provide? This needs to be briefly addressed somewhere in the manuscript.

Reviewer#1

In the submitted manuscript titled 'Metabolic and Circadian Inputs Encode Anticipatory Biogenesis of Hepatic Fed MicroRNAs' by Sandra et al, the authors have followed the initial finding reported earlier by the same group (Maniyadath et al., 2019) to identify the oscillatory expression of certain miRNAs that are known to be controller of important starvation responsive genes. The investigation reported here, they have concluded an interesting hypothesis explaining the observation. They have scored the anticipatory high expression of pri and pre-miRNA of corresponding mature miRNAs that are shown to be down regulated in the process of starvation. This hypothesis has been tested in the context of day light cycle and circadian rhythm and in the context of hypoglycemic and hyperglycemic conditions induced by pharmacological agents. I appreciate the approaches the authors have taken but I am disappointed of not finding any "real" mechanistic insight at molecular level that can explain these observations. The key concerns I have:

1. How the author can be certain that the upregulation (anticipatory) of pr- and pre-miRNAs are happening at post-transcription level and not due to a processing defect caused by the lowering of some components of miRNA machineries that are known to be regulated (like Dicer, or DGCR8) under different metabolic context. The expression of these components should be checked.

Response: We thank the reviewer for this critical input. We have addressed this concern by quantifying the expression of DROSHA and DICER proteins in fed-fast-refed paradigms as suggested by the reviewer. These results have been incorporated in the revised manuscript (Fig 4B-C and Fig S4B-C). To elaborate, while the levels of DROSHA did not alter, we observed a starvation-dependent decrease and a refeed-associated increase in the levels of DICER. The elevated DICER levels in the refeed state correlate well with the upregulation of mature miRs during refeeding and posit a metabolic control of microRNA maturation via DICER.

2. To confirm the regulation happening at transcriptional level, the authors must have done experiments by doing DNA CHIP assay to confirm the enhanced association of the polymerase with the promoter region of the responsive miRNA expressing locus in starvation. It is also

important to have control with the miRNA gene locus that are not expressing in the same manner during starvation-fed cycle.

Response: We agree with the reviewer's comment, and it is indeed important to assay for the Pol II (RNA Polymerase II) association on mir genes in fed and starved conditions. As the abundance of Pol II on promoters does not always correlate with transcription, we checked the levels of elongating, serine-2 phosphorylated Pol II (S2P-Pol II) on miR gene bodies (Fig 4A and Fig S4A). We observed an increased association of S2P-Pol II on hepatic fed miRs Let-7i, miR-204, and miR-221/222 in starvation when compared to a fed state. As the reviewer rightly pointed out, we have used a control miR (miR-99b) which we have earlier found to be unaltered in fed-fast cycles (Maniyadath *et al*, 2019). Consistent with this we now show that there is no change in transcription of miR-99b, as assayed by S2P-Pol II ChIP, unlike the anticipatory fed miRs.

3. The extracellular export of miRNA can also contribute to the level of changed miRNA. It will be wise to look at the circulatory level of target miRNAs, pre-and pri-miRNAs levels in the serum of the animals going through fed and starvation cycle.

Response: We thank the reviewer for this critical input. We checked the levels of pri- pre- and mature miRs in the serum of mice under fed and fasted conditions. While we could not detect pri- and pre-miRs in the serum most likely owing to their absence/low abundance, we could detect mature miRs in circulation. All the miRs in the serum exhibited the same oscillatory pattern as observed in the liver (Fig 4E). This result confirms that the changes in the levels of mature miRs observed in the liver during fed-fast cycles are not due to differential extracellular export.

4. The anticipatory differential expression of mature and precursor levels of miRNAs at the global level should be wisely followed in an bioinformatic approach involving gene expression data from normal fed and starved animal liver. This is essential to find a true link related to metabolic dysregulation and conciliatory miRNA expression-the basis of the hypothesis.

Response: We thank the reviewer for highlighting this. But we respectfully disagree that this aspect should be included in the current manuscript for the following reasons

1. At the outset we would like to point out that in our published work (Maniyadath *et al.*, 2019), we have exhaustively characterized the target mRNAs of these miRs and how they correlate with mRNA and protein expression of pathways relevant to fed-fast-refed cycles. In addition to extensive bioinformatic analysis, this study also provided a mechanistic basis for miR-mediated downregulation of genes that lead to metabolic deficits.
2. The focus of the current manuscript was to unravel the anticipatory biogenesis of the hepatic-fed microRNAs and to uncover the upstream regulators which is a follow-up of the previous study that contains the kind of analysis the referee is pointing towards.
3. Furthermore, in the current manuscript, we have checked the levels of the prominent target genes (starvation transcripts) in *ad libitum* fed, light-fed, and dark-fed groups (Fig S1C and Fig 2D) and observed an anti-correlated pattern with respect to the mature miRs consistent with our earlier publication.

Reviewer #2

(Comments to the Authors (Required)):

Anticipatory behavior and physiological response to food is particularly important for animals to restore metabolic homeostasis promptly and increase the survival/fitness. miRNAs have been identified as important posttranscriptional regulators of gene expressions including hepatic metabolism. In this manuscript, the authors report several hepatic miRNAs that respond to fed at the biogenesis level.

Overall this is an interesting paper mainly focusing on the biogenesis of miRNAs responding to metabolic stages. The findings are interesting and the current evidence seem solid.

We thank the reviewer for the positive comments and we are encouraged that the results presented in the manuscript were found to be of interest.

However, I do have a few things that need to be addressed.

1. In the abstract, the authors mentioned that "we report anticipatory biogenesis of oscillatory hepatic microRNAs that peak during a fed state and inhibit starvation responsive genes". This might have been reported by the authors and others in previous papers, however, I don't see any

"inhibit starvation responsive gene" expression data throughout the current manuscript. Indeed, this is also one of the major caveats for this manuscript. The authors systematically checked the biogenesis of the four miRNAs under different conditions, however there is almost no data of their effects on potential targets. I suggest the authors used their samples to run a few qPCR or even WB to check several potential miRNA targets that might be starvation related here.

Response: We respectfully submit that the reviewer has possibly missed the qPCRs done for target mRNAs which were included in the original submission. Nonetheless, the data presented in Fig SIC and Fig 2D provide an exhaustive characterization of target mRNA expression which is consistent with our published work (Maniyadath *et al.*, 2019) and clearly demonstrate an anti-correlated pattern with respect to the mature miRs.

2. This manuscript lacks explanations of the rationale to study these four oscillatory miRNAs in the beginning of their results. Is this from a screen? Why do they start from these four? If just because they identified these four miRNAs in the previous study, then why they want to do it again? What new does this study provide? This needs to be briefly addressed somewhere in the manuscript.

Response: We agree with the reviewer that the original version of the manuscript lacked a clear explanation in this regard. We have now addressed this concern in the revised manuscript at the beginning of the results section.

References

Maniyadath B, Chattopadhyay T, Verma S, Kumari S, Kulkarni P, Banerjee K, Lazarus A, Kokane SS, Shetty T, Anamika K *et al* (2019) Loss of Hepatic Oscillatory Fed microRNAs Abrogates Refed Transition and Causes Liver Dysfunctions. *Cell Rep* 26: 2212-2226.e2217

February 6, 2024

RE: Life Science Alliance Manuscript #LSA-2023-02180-TR

Prof. Ullas Kolthur-Seetharam
Tata Institute of Fundamental Research
Dept of Biological Sciences
Homi Bhabha Road, Colaba
Mumbai, Maharashtra 400005
India

Dear Dr. Kolthur-Seetharam,

Thank you for submitting your revised manuscript entitled "Metabolic and Circadian Inputs Encode Anticipatory Biogenesis of Hepatic Fed MicroRNAs". We would be happy to publish your paper in Life Science Alliance pending final revisions necessary to meet our formatting guidelines.

- please be sure that the authorship listing and order is correct
- please add ORCID ID for the corresponding author -- you should have received instructions on how to do so
- it looks like there is a discrepancy in the presentation of the name of one of your co-authors: U.S. Sandra in the manuscript file vs. Sandra US in the system-please correct
- the full name (middle name as initials) of each author should be given on the title page
- there is only one panel in Figure S2, so there is no need to label it as A. Please correct the figure and its legend and call out in the manuscript text accordingly
- we encourage you to revise the figure legend for Figure 5 such that the figure panels are introduced in an alphabetical order

A. FINAL FILES:

B. MANUSCRIPT ORGANIZATION AND FORMATTING:

Sincerely,

Reviewer #1 (Comments to the Authors (Required)):

The authors have critically and experimentally addressed all my concerns and I recommend acceptance of this revised manuscript for publication in LSA. Congratulations to the authors.

February 13, 2024

RE: Life Science Alliance Manuscript #LSA-2023-02180-TRR

Prof. Ullas Kolthur-Seetharam
Tata Institute of Fundamental Research
Dept of Biological Sciences
Homi Bhabha Road, Colaba
Mumbai, Maharashtra 400005
India

Dear Dr. Kolthur-Seetharam,

Thank you for submitting your Research Article entitled "Metabolic and Circadian Inputs Encode Anticipatory Biogenesis of Hepatic Fed MicroRNAs". It is a pleasure to let you know that your manuscript is now accepted for publication in Life Science Alliance. Congratulations on this interesting work.

DISTRIBUTION OF MATERIALS:

Again, congratulations on a very nice paper. I hope you found the review process to be constructive and are pleased with how the manuscript was handled editorially. We look forward to future exciting submissions from your lab.

Sincerely,
